# Facile Electrodeposition of NiCo_2_O_4_ Nanosheets on Porous Carbonized Wood for Wood-Derived Asymmetric Supercapacitors

**DOI:** 10.3390/polym14132521

**Published:** 2022-06-21

**Authors:** Jingjiang Yang, Huiling Li, Shuijian He, Haijuan Du, Kunming Liu, Chunmei Zhang, Shaohua Jiang

**Affiliations:** 1Jiangsu Co-Innovation Center of Efficient Processing and Utilization of Forest Resources, International Innovation Center for Forest Chemicals and Materials, Nanjing Forestry University, Nanjing 210037, China; yjj10101010@163.com (J.Y.); yyb23232323@163.com (H.L.); 2College of Textiles, Zhongyuan University of Technology, Zhengzhou 450007, China; duhaijuan2009@126.com; 3Faculty of Materials Metallurgy and Chemistry, Jiangxi University of Science and Technology, Ganzhou 341000, China; liukunming@jxust.edu.cn; 4Institute of Materials Science and Devices, School of Materials Science and Engineering, Suzhou University of Science and Technology, Suzhou 215009, China

**Keywords:** supercapacitor, electrodeposition, electrochemical performance, NiCo_2_O_4_, carbonized wood

## Abstract

Multichannel-porous carbon derived from wood can serve as a conductive substrate for fast charge transfer and ion diffusion, supporting the high-theory capacitance of pseudocapacitive materials. Herein, NiCo_2_O_4_ nanosheets, which are hierarchically porous, anchored on the surface of carbonized wood via electrodeposition for free-binder high-performance supercapacitor electrode materials, were proposed. Benefiting from the effectively alleviated NiCo_2_O_4_ nanosheets accumulation and sufficient active surface area for redox reaction, a N-doped wood-derived porous carbon-NiCo_2_O_4_ nanosheet hybrid material (NCNS–NCW) electrode exhibited a specific electric capacity of 1730 F g^−1^ at 1 A g^−1^ in 1 mol L^−1^ KOH and splendid electrochemical firmness with 80% capacitance retention after cycles. Furthermore, an all-wood-based asymmetric supercapacitor based on NCNS–NCW//NCW was assembled and a high energy density of 56.1 Wh kg^−1^ at a watt density of 349 W kg^−1^ was achieved. Due to the great electrochemical performance of NCNS–NCW, we expect it to be used as an electrode material with great promise for energy storage equipment.

## 1. Introduction

Supercapacitors are green and sustainable devices for energy storage, with long service life, high power density, and fast charge and discharge, which have attracted widespread attention [1,2,3,4,5,6]. In order to improve the electrochemical performance of supercapacitors, the most critical factor is the electrode material [7,8]. Porous carbon materials [8,9,10,11,12,13] have long been widely regarded as the ideal electrode materials for supercapacitors because of their superior stability, controllable high-electrical conductivity, and pore structure [14,15]. On the other hand, pseudocapacitive electrode materials, based on hydroxides, with high-theory-specific capacitance transitional metal oxides (TMOs), have been frequently investigated in recent years [16,17]. Among these metal compounds, NiCo_2_O_4_ with abundant redox couples (Ni^2+^/Ni^3+^ and Co^2+^/Co^3+^) can provide high-theoretical capacitance (2178 F g^−1^ or 334 mAh g^−1^), thereby attracting much attention [18,19]. However, NiCo_2_O_4_ suffers from poor electrical conductivity and structural instability during charging and discharging, which negatively affects its electrochemical performance. [20,21]. Thus, it is feasible to design porous carbon material-composited NiCo_2_O_4_ electrode materials with a special structure, which could combine high conductivity with high-specific capacitance and avoid their weaknesses [20,22,23]. For example, NiCo_2_O_4_@CNT electrode materials could be successfully fabricated, where CNT films serve as a conductive template and NiCo_2_O_4_ nanosheets provide more reaction-active sites [24,25]. Therefore, suitable conductive scaffolds for supporting NiCo_2_O_4_ are significant for high-performance supercapacitors [26,27].

Wood, a natural and renewable resource from trees, possesses abundant porous structure and aligned microchannels for the penetration and diffusion of ions [28,29,30,31,32]. Carbonized wood can inherit original three-dimensional (3D) structures, which is considered as an ideal supporter for loading metal compounds and hold tremendous potential for applications in energy storage [33,34,35]. Recently, researchers have demonstrated that it is practicable to raise the electrochemical performance of such a hybrid electrode via combining heteroatom doping for carbon substrates. Many studies have demonstrated that the electrochemical performance of porous carbon electrodes can be enhanced by heteroatom doping/hybridizing with TMOs. Zhang et al. [36] pyrolyzed natural balsa wood to obtain a wood-based carbon electrode matrix and then deposited MnO_2_ and graphene–carbon quantum dots on its surface using the hydrothermal method. MnO_2_ and graphene–carbon quantum dots form unique needle-like nanostructures on the carbon matrix surface, which can act as a 3D pathway, enabling electron transport and electrolyte penetration/diffusion rates to be accelerated. After optimization, the mass specific capacitance of the electrode reaches 188.4 F g^−1^. However, due to the redox reaction containing the entire charge storage process of this capacitor material, the electrode material undergoes an irreversible phase transition, resulting in poor cycle performance from the electrode material. Tang et al. [37] optimized the porous structure and specific surface area of wood, followed by doping N atoms and foaming agents during pyrolysis, and the prepared TARC samples were activated in an ammonia flow (TARC-N). The electrochemical characterization results showed that the specific capacitance of TARC-N could reach 704 F g^−1^. However, due to the inability to form a good chemical bond between the dopant and the substrate in the above method, the low amount of heteroatom doping greatly reduces the electrochemical performance of the wood electrode. Recently, our group applied a hydrothermal reaction to deposit high-loading MnO_2_ nanosheets to wood-derived carbon as an electrode for supercapacitors [38]. The electrode presented high-rate areal/specific capacitance (current density: 20 mA cm^–2^) and the assembled symmetric supercapacitor also possessed 75.2% capacitance retention after 10,000 long-term cycles (current density: 20 mA cm^–2^) and high energy densities (0.502 mWh cm^–2^/12.2 Wh kg^–1^). Thus, we focus on the development of N-doped wood-derived porous carbon-NiCo_2_O_4_ nanosheet hybrid materials (NiCo_2_O_4_-NCW, NCNS–NCW) by electrodeposition technology, building brief spread distance to enhance the stability and capacitance, so that electrolyte ions and electron transfer can spread faster. The resultant NCNS–NCW showed a high specific capacitance (1730 F g^−1^ at 1 A g^−1^). The resultant NCNS–NCW showed excellent cycle life (92.5% capacitance preservation in the back of turns at 10 A g^−1^). In addition, the device based on NCNS–NCW//NCW exhibited a stable voltage window from 0 V to 1.6 V, and maximal energy was up to 56.1 Wh kg^−1^ at a power density of 349 W kg^−1^. The amazing consequences prove that Ni-Co oxide nanosheets on wood-derived carbon with different structures are an efficient method in order to reinforce the electrochemical behavior of electrodes.

## 2. Materials and Methods

### 2.1. Preparation of NCW

The camellia tree trunk (collected from Hunan province, China.) was cut perpendicularly to the direction of growth and put into 1.0 mol L^−1^ NH_4_Cl solution for 4 h. The treated wood was put into the vacuum drying cabinet for 6 h at 60 °C, and then metastasized into a tube furnace carbonized at 800 °C for 2 h at 5 °C min^−1^ under N_2_ flowing.

### 2.2. Preparation of NCNS–NCW

The preparation process for the porous carbon-NiCo_2_O_4_ derived from wood nanosheet hybrid materials (NCNS–NCW) was carried out through a facile electrodeposition technology. This technology was combined with a post-heating procedure. Firstly, working electrodes of N-doped wood carbon were immersed into a mixed solution containing 0.005 mol L^−1^ Ni(NO_3_)_2_·6H_2_O, 0.01 mol L^−1^ Co(NO_3_)_2_·6H_2_O, and 0.1 mol L^−1^ Na_2_SO_4_. Ag/AgCl and Pt foil, respectively, served as the reference and counter. The electro-synthesis process was carried out at different periods of scanning rate 5 mV s^−1^ and voltage range −1.2 V to 0.2 V. After the deposition, we washed the electrodes with ionized water, then dried the electrodes in a vacuum oven at 60 °C for 12 h. Ultimately, to improve the crystallinity of the product, the electrode was annealed at 350 °C with a slow ramp rate of 3 °C min^−1^ in a N_2_ atmosphere for 90 min. The quality of the deposited NiCo_2_O_4_ nanosheets was measured from the mass difference before and electrochemical deposition and after post-heating treatment. The test results showed that the average weight of NCW was 10.0 mg and the average weight of deposited nanosheets was 1.0, 2.0 and 3.0 mg, respectively. The weights of NCW and the deposited NiCo_2_O_4_ nanosheets (NiCo_2_O_4_NSs) were listed in Appendix A. The NCW and NCNS–NCW materials directly utilized as self-standing electrodes after washing with alcohol under supersonic treatment. The whole preparation process of NCW@NiCo_2_O_4_ was shown in Appendix A.

### 2.3. Physicochemical Characterization

To observe the morphology of the samples, we used transmission electron microscopy (TEM, JEM-1400) and scanning electron microscopy (SEM, S-4800) to observe. The crystal framework was identified by X-ray diffractometer (XRD, BukerD8Phaser) with Cu Kα irradiation (λ = 1.54184 Å). Raman spectra were measured with a Raman microspectrometer (Jobin Yvon HR800, Edison, NJ, USA) and the wavelength of laser was 633 nm. The desorption isotherms and nitrogen adsorption were obtained using the Micrometrics ASAP 2020 analyzer. The Brunauer–Emmett–Teller (BET) and Barrett–Joyner–Halenda (BJH) methods were, respectively, used to calculate the pore size allocation and specific surface area. X-ray photoelectron spectroscopy (XPS) analyses were put into effect by using a Perkin-Elmer PHI 550 (Thermo Fisher Scientific, Waltham, MA, USA) at ultrahigh vacuum using Al Kα X-rays source.

### 2.4. Supercapacitor Measurements

A series of electrochemical tests such as galvanostatic charge–discharge (GCD), cyclic voltammetry (CV), and electrochemical impedance spectroscopy (EIS) were performed using the CHI760e electrochemical workstation (Chenhua, Shanghai, China). The NCW and NTNS-NCW were used as working electrodes in three-electrode system; the platinum foil electrode and Hg/HgO electrode were used as counters and reference electrodes, respectively. The test of the device is similar to the three-electrode test, using NCNS–NCW as the positive electrode and NCW as the negative electrode, in KOH electrolyte. The device for the electrochemical performance test can be found in Appendix A. EIS measurements were measured in the open-circuit potential range of 0.01~105 Hz with an abundance of 5 mV. We calculated the specific capacitance (C_s_ in F g^−1^) according to the constant current charge–discharge profile, and the formula is as follows:(1)$$C_{s}=\frac{I\times \Delta t}{m\times \Delta V}$$
where i (mA) is the discharge current, m (mg) represents the quality of electrode material (above 1 mg), and ΔV (V) is the potential window Δt (s) within the discharge time.

The energy density (E, Wh kg^−1^) is calculated from the discharge distribution of the two-electrode system with the following formula:(2)$$E=\frac{1}{2}\times C_{s}\times \Delta V^{2}$$
where E (Wh kg^−1^) is the energy density, Cs (F g^−1^) is the specific capacitance of the active material, and ΔV (V) is the discharge voltage range.

The power density (P) is calculated as:(3)$$P=\frac{E}{\Delta t}\;$$
where P (kW kg**^−^**^1^) is the power density, E (Wh kg**^−^**^1^) is the energy density, and Δt (s) is the discharge time.

## 3. Results

The surface appearance of the prepared objects was firstly characterized to confirm the unique 3D hierarchical porous structure of NCW and NCNS–NCW. Looking at Figure 1, it can be found that the wood-derived N-doped carbon (NCW) exhibited multiple well-aligned porous channels, 5–10 and 40–60 μm in diameter (Figure 1a,b). The NiCo_2_O_4_ nanosheets were uniformly and vertically grown on the surface of NCW for NCNS–NCW (Figure 1c,d) and the nanosheets interlace and form a 3D network structure. Figure 1d showed that the width and thickness of the NiCo_2_O_4_ nanosheets, respectively, were 200~500 nm and 5~8 nm. TEM and HRTEM were employed to further provide clearer information about the microstructure features of the NCNS–NCW. Figure 1d,e show that the length of the NiCo_2_O_4_ nanosheet was about 200 nm. The lattice fringe of the NiCo_2_O_4_ nanosheet shown in Figure 1f is about 0.25 nm, corresponding to the (311) and (220) planes of the NiCo_2_O_4_ phase, respectively. Fabrication of the binder-free electrode materials with vertical nanoarchitecture is the prominent approach to achieve exalted energy storage performance. Figure 1g shows the EDS elemental mapping images of the NCW@NiCo_2_O_4_-2, indicating the existence of C, N, O, Co, and Ni elements in the samples.

The chemical composition and crystal properties of the obtained NCW and NCNS–NCW composites can be confirmed from the XRD test results (Figure 2). As presented in Figure 2a, two peaks present in NCW well matched the (002) and (100) planes of amorphous carbon. For NCNS–NCW (Figure 2b), the characteristic diffraction peaks of 2θ = 31.2°, 36.8°, 44.7°, 59.2°, and 65.1° have a good exponential relationship with (220), (311), (400), (511), and (440) surfaces of the spinel NiCo_2_O_4_ phase (JCPDSNo.73-1702) [39,40,41].

The chemical bonding states of NCNS–NCW were studied using XPS analysis. High-resolution XPS spectra of Ni 2p, Co 2p, and O 1s are correspondingly shown in Figure 2e,f. NCNS–NCW’s high-resolution Co 2p and Ni 2p spectra match best with two spin–orbit bimodals and two shakeup satellites (identified as “Sat.”). Two spin–orbital peaks can be used to fit the Ni 2P spectrum (Figure 2d). The two peaks are Ni 2p3/2 (853.2 eV) and Ni 2P1/2 (872.4 eV), respectively. Each peak is attended by a different shaking satellites signal (situated at 860.5 and 879.0 eV, labeled “Sat”) [19,42]. The spin–orbit binding energies at 854.4 and 871.9 eV are Ni^3+^ 2p^3/2^, and Ni^3+^ 2p^1/2^, respectively, and 855.1 and 873.4 eV are Ni^2+^ 2p^3/2^ and Ni^2+^ 2p^1/2^, respectively. The spectra and matching results of Co 2p (Figure 2e) are very similar to those of Ni 2p. As shown in Figure 2e, the Co 2p spectra agree well with Co 2p_3/2_ of 779.9 eV, Co 2p_1/2_ of 795.4 eV, and two oscillating satellites 784.3 and 802.0 eV, respectively [43]. The double peaks at 779.7 and 795.1 eV belong to Co^3+^ 2p^3/2^ and Co^3+^ 2p^1/2^, and the other pair at 780.5 and 796.8 eV correspond to Co^2+^ 2p^3/2^ and Co^2+^ 2p^1/2^. The binding energies of Ni 2p and Co 2p are consistent with the results measured in the literature [18,44]. In the area of O 1s spectrum (Figure 2f), oxygen in the air reacts with the metal–oxygen bond in NiCo_2_O_4_ with peaks of 529.5 eV and 530.7 eV, respectively [45,46]. In consequence, XPS results confirm the presence of Ni^2+^, Ni^3+^, Co^3+^, Co^2+^, and O^2−^, which was consistent with the phase analysis of the NNCS–NW-2 composite.

The highly conductive NCW substrate provides an adequate pathway for electron transport and the vertical NiCo_2_O_4_ nanosheet provides active sites for numerous reversible redox reactions. In the meantime, the large number of pores between NiCo_2_O_4_ nanosheets provides sufficient conditions for the entry of electrolyte ions, which result in an improvement in its electrochemical properties [47,48]. As can be seen from Appendix A in the Appendix A, EIS measurements were further performed on the NCW@NiCo_2_O_4_ composite electrode in a frequency range of 100 kHz–0.01 Hz. NCW can serve as a well-supported conductive substrate for NiCo_2_O_4_NSs. Therefore, the low R_s_ value (2.57 Ω) and R_ct_ value (1.3 Ω) of the NCW@NiCo_2_O_4_-2 electrode suggest that the electrode has lower bulk resistance, faster electron transfer kinetics, and lower charge diffusion resistance. R_s_ value means series resistance and R_ct_ value means charge transfer resistance. The lower resistance of NCW@NiCo_2_O_4_-2 is due to the 3D porous interconnected conductive structure, which provides more exposed active centers and fast transfer paths for ions and electrons. Figure 3a,b display the CV data at 10~50 Mv s^−1^ and GCD data at 1~20 A g^−1^ of NCW electrodes. Even at high scan rates, the CV curve of the NCW clearly shows a quasi-rectangular shape and the GCD plot is a roughly symmetrical triangle, with the NCW electrode showing excellent electrochemical charge–discharge performance. Figure 3c,d display the CV data at 10~5 mV s^−1^ and GCD data at 1~10 A g^−1^ of NCNS–NCW electrodes. Two pairs of redox peaks appear in NNCS–NCW due to the reversible redox reactions of Ni^2+^/Ni^3+^ and Co^2+^/Co^3+^. The reaction is described as follows [49,50]:(4)$${\mathrm{NiCo}}_{2}O_{4}+{\mathrm{OH}}^{-}+H_{2}O\;↔\;\mathrm{NiOOH}+\mathrm{2CoOOH}+e^{-}$$
(5)$$\mathrm{CoOOH}+{\mathrm{OH}}^{-}\;↔\;{\mathrm{CoO}}_{2}+H_{2}+e^{-}$$

The specific capacities of NCNS–NCW and NCW were further calculated from GCD curves at 1 A g^−1^. As exhibited in Figure 3b,d, the consistent discharge time, respectively, was 71.7 s, 110.2 s, and 236 s.

Because the NNCS–NCW electrode possesses a high specific discharge capacity, its specific discharge capacity is 1730 F g^−1^ at 1 A g^−1^, and the specific discharge capacity retention rate of the NCNS–NCW electrode could be as high as 46.18%, at a high current density of 10 A g^−1^, verifying its ideal rate performance.

Significant nonlinear discharge plateaus and redox peaks can be found by observational analysis of the CV and GCD curves, which reflected the battery characteristics of the prepared NNCS–NCW electrode. Furthermore, the capacity contribution and kinetic origin were verified. According to the power law equation, capacitance effects can be qualitatively calculated by studying the relationship between the peak redox current (i) and the corresponding scan rates (*v*) [50,51]:i = *a**v*^b^(6)
where *a* and b are constants. The slope is b, calculated by the following formula:logi = log*a* + blog*v*(7)

It can be seen from Figure 3e that log *v* depends on log i, and the b values of each peak are 0.84, which is a process dominated by surface capacitance [40,52,53]. The current value I at stable potential V can be divided into capacitive effect (k_1_*v*) and diffusion-controlled contribution (k_2_*v*^1/2^), so the ratio of capacitance effect can be obtained by the following formula:i(V) = k_1_*v* + k_2_*v*^1/2^(8)

Equation (8) can be further verified from:i(V)/*v*^1/2^ = k_1_*v*^1/2^ + k_2_(9)

Figure 3f was obtained by fitting the capacitance subscription current designed by the k_1_ value, reflecting that the capacitance effect accounted for 72.9% of the total capacity. As is known, as the scanning rate enhances, the diffusion control process is inhibited and the capacitance contribution increases.

Upon optimizing the quality of both negative and positive electrodes, an ASC was assembled for evaluating the practical application, where the NCW was employed as the negative electrode and NCNS–NCW was applied as the positive electrode. Figure 4a,b show the CV curves and charge–discharge curves of NCNS–NCW//NCW in different voltage ranges, respectively. From the CV curves, a voltage window of 1.6 V causes polarization. Furthermore, the potential window of GCD curves increases from 1.1 V to 1.5 V. Therefore, the stable operational voltage of the obtained ACS ranges from 0 V to 1.5 V. Figure 4c delivered CV curves for the ACS at individual scanning rates. At a high scanning rate of 100 mV s^−1^, the shape of the CV curve changes very little, indicating that NTNS–NCW has good electrochemical reaction rate and reversibility. Under the voltage window of 0~1.5 V, when the instrument is at different current densities, the measured GCD curve is a triangular curve with no visible plateau (Figure 4d), implying a highly reversible redox reaction in the device. According to the GCD curve, when the current density is 1, 2, and 20 A g^−1^, the specific capacitors of the device are 119, 108, 93, and 72 F g^−1^, respectively. Obviously, 61% of the initial specific capacity can be retained when the current density was high at 20 F g^−1^, indicating good rate performance of the ASC. These results further demonstrated that the prepared NiCo_2_O_4_ material possessed superior excellent rate capability and high capacitance, which can provide potential promise for real applications. The electrochemical impedance spectra (EIS) of NNCS–NCW and distance (R_ct_) are minimal, and the minimum linear slope per electrode in the low-frequency region, and the NCNS–NCW electrode, have maximum inherent ohmic resistance of 0.72 Ω, and the linear slope is the largest. The conflict in the slope of the straight line of NCNS–NCW and NCW electrodes derived from their different shapes and sizes. Vertical 2D NiCo_2_O_4_ nanosheets grown on NCW substrates provided sufficient channels for the spread of ions, thus, reducing spread resistance. Therefore, combined with many active sites for invertible redox reactions from NiCo_2_O_4_ nanosheets and the low R_s_ and R_ct_ from the NCW electrode, the NCNS–NCW electrode achieved the fastest electron and ion transfer rates, thus, significantly improving the electrical conductivity and electrochemical activity. On the other hand, energy density (E) and power density (p) were also two essential factors in the supercapacitor device for practical application, described here by the Ragone plots and shown in Figure 4f. Competing with the NiCoO_2_-based systems reported in recent years, when the power density is 800.2 W kg^−1^, the energy density of NTNS–NCW//NCW can be 56.1 Wh kg^−1^, and when the power density is 800.2 Wkg^−1^, the energy density can still be 16.4 Wh kg^−1^, such as NiCo_2_O_4_/GCNF//AC (48.6 Wh kg^−1^ at 749.3 W kg^−1^) [54], WC/CuO//WC (13.6 Wh kg^−1^ at 350.3 W kg^−1^) [55], and P-NCO NWs/NF//RGO (53.39 Wh kg^–1^ at 779.82 W kg^–1^) [56]. As shown in Appendix A, the assembled symmetric supercapacitor was also found to have 75.2% capacitance retention and high energy density (0.502 mWh cm^–2^/12.2 Wh kg^–1^) by testing the assembled symmetric supercapacitor under 10,000 long-term cycling conditions.

## 4. Conclusions

Wood-based carbon with a unique microchannel structure and hierarchical porosity was considered as a renewable resource for applications in energy storage. Herein, hierarchical porous NiCo_2_O_4_-NCW composites were successfully prepared by electrodeposition technology. NiCo_2_O_4_–NCW (NCNS–NCW) integrated the full advantages of different components and exhibited high specific capacitance of 1730 F g^−1^. Furthermore, an all-wood-based asymmetric supercapacitor was assembled by employing NCNS–NCW and NCW. The maximum specific energy of positive and negative electrodes was 56.1 Wh kg^−1^ at 349 kW kg^−1^. Such an excellent electrochemical performance of NCNS–NCW and NCW is basically attributed to the low resistance of the free binder electrode, which effectively alleviates the accumulation of NiCo_2_O_4_ nanosheets and has sufficient active surface area for redox reactions. Therefore, the synthesized layered NCNS–NCW composites can be used as potential electrode materials for advanced energy storage devices.

## Figures and Tables

**Figure 1 polymers-14-02521-f001:**
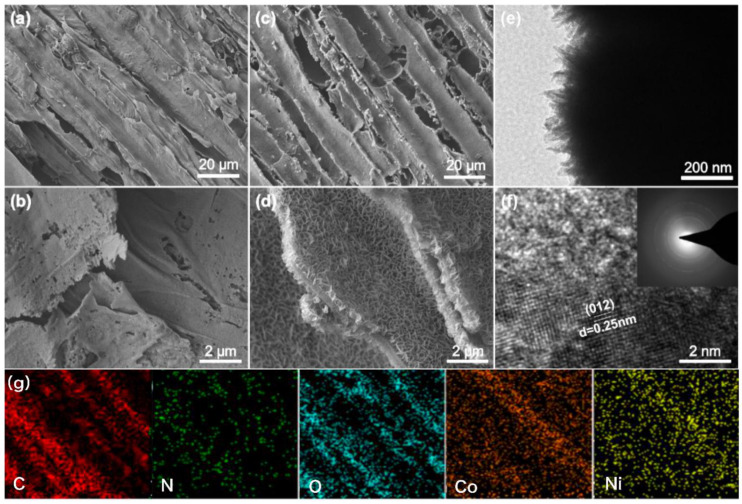
SEM images of NCW (**a**,**b**) and NCNS–NCW (**c**,**d**); TEM (**e**) and HRTEM (**f**) images of NCNS–NCW. (**g**) EDS elemental mapping images of the NCW@NiCo_2_O_4_-2.

**Figure 2 polymers-14-02521-f002:**
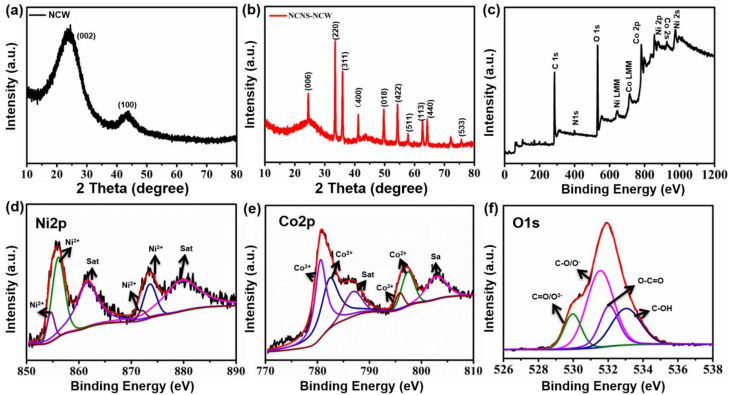
(**a**) XRD patterns of NCW and NCNS−NCW. (**b**) XPS of NCNS−NCW. (**c**) Survey scan, (**d**) Ni 2p, (**e**) Co 2p, (**f**) O 1s of the sample.

**Figure 3 polymers-14-02521-f003:**
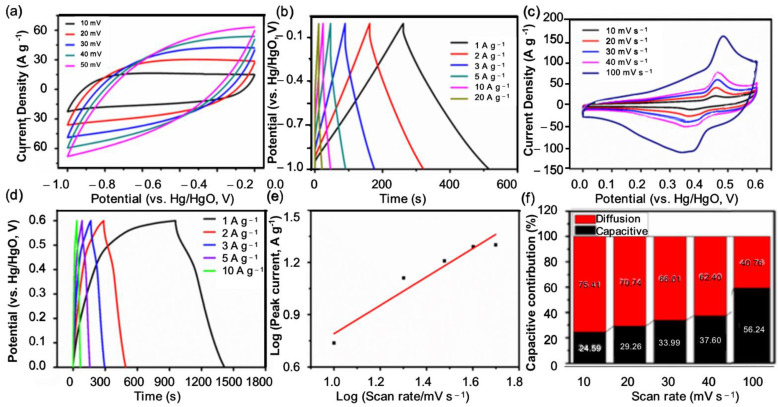
CV (**a**) and GCD (**b**) curves of NCW; CV (**c**) and GCD (**d**) curves of NCNS–NCW; (**e**) square root of scan rate and current density of NCNS–NCW; and (**f**) standardized contribution ratio of capacitance capacity at different scanning rates.

**Figure 4 polymers-14-02521-f004:**
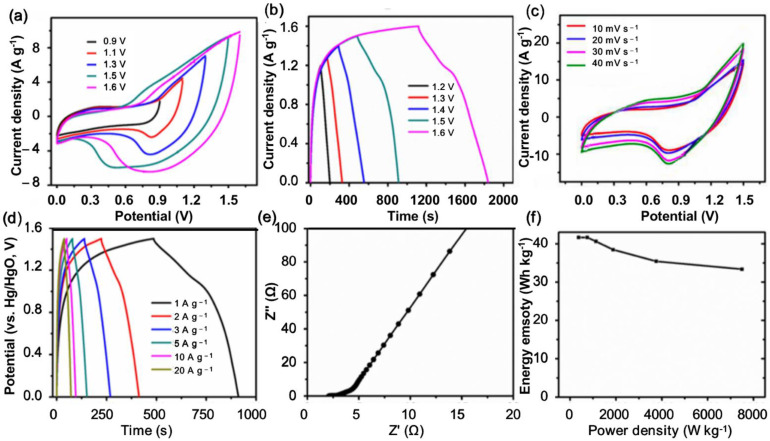
(**a**) CV curve of asymmetric supercapacitor apparatus at different window potential; (**b**) GCD profile of ASC at various cell voltages (1 A/g); (**c**) CV curves of ASC at different scanning rates; (**d**) GCD curve of ASC at different current densities; (**e**) EIS spectrum of fabricated ASC device; (**f**) power density–energy density relationship of a prepared asymmetric supercapacitor (Ragone diagram).

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
