# Peer review of "Facile Electrodeposition of NiCo2O4 Nanosheets on Porous Carbonized Wood for Wood-Derived Asymmetric Supercapacitors"

_polymers, 2022, doi:10.3390/polym14132521_

Round 1

Reviewer 1 Report

The paper needs modifcation. Few suggestions are given below

1: The introduction is very short and general. Please ass some literature on the subject and compare novelty of your work.

2: It will be interesting to see the real pictures of the fabricated electrode. Please add to the experimental section while talking about the electrode fabrication.

3: The authors state their electrode is highly conducting. Did they measure the conductivity or they are just claiming this on the basis of CV curves?

4: Please also add the picture of electrochemical cells containing all electrodes.

Author Response

Response to reviewer #1’s comments

Dear Editors and Reviewers,

Thank you for your letter and for the comments from the reviewers concerning our manuscript entitled “Facile electrodeposition of NiCo2O4 nanosheets on porous carbonized wood for wood-derived asymmetric supercapacitors”. We have carefully studied the opinions of the reviewers and revised the manuscript accordingly. These corrections are highlighted with a red color in the revised manuscript for review.

Reviewer #1:
1. The paper needs modification. Few suggestions are given below.1. The introduction is very short and general. Please add some literature on the subject and compare novelty of your work.

Response: Thanks for your valuable suggestion about the manuscript. We have illustrated the innovation of this work with examples in the introduction part of the manuscript.

Recently, more researches have demonstrated that it is practicable to raise the electro-chemical performance of such a hybrid electrode via combining heteroatoms doping of carbon substrates. Many studies have proved that the electrochemical performance of porous carbon electrode can be improved by heteroatom doping/hybridizing with TMOs. Zhang et al. [36] pyrolyzed natural balsa wood to obtain a wood-based carbon electrode matrix, and then deposited MnO2and graphene-carbon quantum dots on its surface by hydrothermal method. MnO2 and graphene-carbon quantum dots form unique nee-dle-like nanostructures on the carbon matrix surface, which can provide fast 3D pathways for electron transport and electrolyte penetration/diffusion. After optimization, the mass specific capacitance of the electrode reached 188.4 F g-1. However, due to the redox reaction involved in the entire charge storage process of this capacitor material, an irreversible phase transition occurs in the electrode material, resulting in poor cycle performance of the electrode material. Tang et al. [37] optimized the porous structure and specific surface area of wood, followed by doping N atoms and foaming agents during pyrolysis, and the prepared TARC samples were activated in an ammonia flow (TARC-N). The electro-chemical characterization results showed that the specific capacitance of TARC-N could reach 704 F g-1. However, due to the inability to form a good chemical bond between the dopant and the substrate in the above method, the amount of heteroatom doping is low, which greatly reduces the electrochemical performance of the wood-based electrode. Recently, our group applied hydrothermal reaction to deposit high-loading MnO2 nanosheets to wood-derived carbon as electrode for supercapacitors [38]. The electrode presented high-rate areal/specific capacitance (current density: 20 mA cm–2) and the assembled symmetric supercapacitor also possessed high energy densities ( 0.502 mWh cm–2/12.2 Wh kg–1) and 75.2% capacitance retention after 10 000 long-term cycles (current density: 20 mA cm–2).

  1. It will be interesting to see the real pictures of the fabricated electrode. Please add to the experimental section while talking about the electrode fabrication.

Response: Thanks for your valuable suggestion. According to your suggestion, we have added real pictures of the electrodes. As shown in Figure S1. Figure S1 is placed in supplementary information.

  1. The authors state their electrode is highly conducting. Did they measure the conductivity or they are just claiming this on the basis of CV curves?

Response: Thanks for your valuable suggestion about the manuscript. According to your suggestion, we have added the electrochemical impedance spectra of NCW@NiCo2O4 composite electrodes as shown in Figure S2. The corresponding discussion was also added in the revised manuscript. Figure S2 is placed in supplementary information.

As can be seen from the Figure S2 in Supplementary Information, EIS measurements were further performed on the NCW@NiCo2O4 composite electrode in the frequency range of 100 kHz-0.01 Hz. NCW can serve as a well-supported conductive substrate for Ni-Co2O4NSs. Therefore, the low Rs value (2.57 Ω) and Rct value (1.3 Ω) of the NCW@NiCo2O4-2 electrode suggest that the electrode has lower bulk resistance, faster electron transfer kinetics, and lower charge diffusion resistance. Rs value means series resistance and Rct value means charge transfer resistance. The lower resistance of NCW@NiCo2O4-2 is attributed to the three-dimensional porous interconnected conductive structure, which provides more exposed active centers and fast transfer paths for ions and electrons.

  1. Please also add the picture of electrochemical cells containing all electrodes.

Response: Thanks for your valuable suggestion about the manuscript. We have modified the figures to add the picture of electrochemical cells containing all electrodes. The corresponding revision has been added into the manuscript. Figure S3 is placed in supplementary information.

The test of the device is similar to the three-electrode test, using NCNS-NCW as the positive electrode and NCW as the negative electrode, in KOH electrolyte. The schematic is shown in the Figure S3.

Reviewer 2 Report

The manuscript entitled “Facile electrodeposition of NiCo2O4 nanosheets on porous carbonized wood for wood-derived asymmetric supercapacitors“ by Jingjiang Yang et al. reports a study concerning the performance of the supercapacitor electrodes based on NiCo2O4 nanosheets. The discussions are well structured. However, the following changes are necessary to be included in the revised manuscript:

  1. equation 4 must to be rewritten;
  2. a comment concerning the influence of the KOH concentration on energy density and power density of supercapacitors.

I recommend this article to be published in the Polymers journal after major revision

Author Response

Response to reviewer #2’s comments

Dear Editors and Reviewers,

Thank you for your letter and for the comments from the reviewers concerning our manuscript entitled “Facile electrodeposition of NiCo2O4 nanosheets on porous carbonized wood for wood-derived asymmetric supercapacitors”. We have carefully studied the opinions of the reviewers and revised the manuscript accordingly. These corrections are highlighted with a red color in the revised manuscript for review.

Reviewer #2:
The manuscript entitled “Facile electrodeposition of NiCo2O4 nanosheets on porous carbonized wood for wood-derived asymmetric supercapacitors“ by Jingjiang Yang et al. reports a study concerning the performance of the supercapacitor electrodes based on NiCo2O4 nanosheets. The discussions are well structured. However, the following changes are necessary to be included in the revised manuscript:
1. Equation 4 must to be rewritten;

Response: Thanks for your valuable suggestion about the manuscript. We have modified the equation 4 in the manuscript. The revisions have been marked in red color in the revised manuscript.

  1. A comment concerning the influence of the KOH concentration on energy density and power density of supercapacitors.

Response: Thanks for your valuable suggestion about the manuscript. The electrolyte used in the experimental part of this work is 3 mol L-1 KOH, which has been used in many reports. Thank you for the valuable comments again. The effect of electrolyte concentration on the supercapacitor was not investigated and can be investigated in the future.

Reviewer 3 Report

The work is about the fabrication process of NiCo2O4 nanosheets on porous carbonized wood using electrodeposition for wood-derived asymmetric supercapacitors. The manuscript was well written in English, however, some points can be done for improvement:

  1. There are minor errors in grammar and vocabulary. Please double-check the manuscript.
  2. Please show a description of the acronyms, events in the abstract. For example, what is NCW?
  3. In material and methods, please explain how the authors made the carbon working electrode from porous carbon material.
  4. The mass of the deposited nanosheets was measured to be 0.01 mg. Please provide the sensitivity of the used microbalance, since 0.01 looks like the minimum of the equipment, which is the error range. Also, provide the mass of carbon that the nanosheets were attached to.
  5. About the TEM figure. Since it is impossible to distingue the carbon and metal oxide, please insert the elemental mapping figure.
  6. Page 5, line 156. The “put down” word is not suitable in this situation, please choose another word.
  7. For the equations on page 6, line 174, please change the arrows.
  8. In figure 3c, the CV curve at a scan rate of 50 mV/s seems too different from the others, though the 40 and 100 mV/s are similar to each other. Please explain.
  9. Why the authors did not study the window potential of each electrode, but did it with the NTNS-NCW //NCW device (anode and cathode running together)?
  10. Figures 4a, 4c, please update the X-axis. In this case, the authors did not use the reference electrode, so it should not have “ vs. Hg/HgO, V”
  11. Please zoom in on the foot of EIS, we need to be able to see it clearly.
  12. Please measure the EIS of NCW as well as NTNS-NCW.
  13. It was introduced in the introduction that the material can be stable up to 10000 cycles, but we did not see the data about this, please provide it. Also please try to give the reason why the stability of the NTNS-NCW //NCW is smaller, which is only 7000 cycles.

For these reasons, we recommend the manuscript to be accepter after major revision.

Author Response

Response to reviewer #3’s comments

Dear Editors and Reviewers,

Thank you for your letter and for the comments from the reviewers concerning our manuscript entitled “Facile electrodeposition of NiCo2O4 nanosheets on porous carbonized wood for wood-derived asymmetric supercapacitors”. We have carefully studied the opinions of the reviewers and revised the manuscript accordingly. These corrections are highlighted with a red color in the revised manuscript for review.

Reviewer #3:

The work is about the fabrication process of NiCo2O4 nanosheets on porous carbonized wood using electrodeposition for wood-derived asymmetric supercapacitors. The manuscript was well written in English, however, some points can be done for improvement:

  1. There are minor errors in grammar and vocabulary. Please double-check the manuscript.

Response: Thanks for your valuable suggestion about the manuscript. We have corrected the grammatical errors in the text. Deleted “and”at line 35; Remove extra spaces in multiple places in the article; correct the spelling of which in line 15; add “as” in line 24; modify the sentence structure of the first sentence in line 29; use “for” to replace “or” in line 69; change “an” in line 81 to “a”; Modify concentration units; delete “the” in line 86; delete extra period in line 111; replace “keep” with “kept” in line 115; capitalize the first letter of lines 131, 135, and 138; modify sentence tense in line 144; modify line 171 incoherent sentences.

  1. Please show a description of the acronyms, events in the abstract. For example, what is NCW?

Response: Thanks for your valuable suggestion about the manuscript. We have explained NCW and NCNS-NCW in the abstract section. The revisions have been marked in red color in the revised manuscript.

  1. In material and methods, please explain how the authors made the carbon working electrode from porous carbon material.

Response: Thanks for your valuable suggestion about the manuscript. More related details have been added into the revision.

The NCW and NCNS-NCW materials directly utilized as self-standing electrodes after washing with alcohol under supersonic treatment.

  1. The mass of the deposited nanosheets was measured to be 0.01 mg. Please provide the sensitivity of the used microbalance, since 0.01 looks like the minimum of the equipment, which is the error range. Also, provide the mass of carbon that the nanosheets were attached to.

Response: Thanks for your valuable suggestion about the manuscript. The mass of the deposited nanosheets was summarized in Table S1. By controlling the different electrodeposition times of the samples, samples with different loadings can be obtained. In addition, the mass for the NCW and the deposited nanosheets was measured by a micro-balance (Sartorius BT125D) with an accuracy of 0.01 mg, which can guarantee the accuracy of the data. The specific load is shown in the Table S1.

Tab. S1 Mass loading of active material.

Sample

NCW (mg)

NiCo2O4 NSs (mg)

NCW

1.0

0

NCW@NiCo2O4-1

1.0

1

NCW@NiCo2O4-2

1.0

2

NCW@NiCo2O4-3

1.0

3

The mass of the deposited NiCo2O4 nanosheets was measured from the mass difference before and electrochemical deposition after post-heating treatment by means of a micro-balance (Sartorius BT125D) with an accuracy of 0.01 mg.

  1. About the TEM figure. Since it is impossible to distingue the carbon and metal oxide, please insert the elemental mapping figure.

Response: Thanks for your valuable suggestion about the manuscript. The elemental mapping figure is as follows in Figure S4. Figure S4 is placed in supplementary information.

  1. Page 5, line 156. The “put down” word is not suitable in this situation, please choose another word.

Response: Thanks for your valuable suggestion about the manuscript. The word “put down” has been corrected into “corresponded”.

  1. For the equations on page 6, line 174, please change the arrows.

Response: Thanks for your valuable suggestion about the manuscript. The canonical use of arrows has been revised in the text and marked in red.

  1. In figure 3c, the CV curve at a scan rate of 50 mV/s seems too different from the others, though the 40 and 100 mV/s are similar to each other. Please explain.

Response: Thanks for your valuable suggestion about the manuscript. There is a difference in the 40 mv s-1 and 100 mv s-1 curve. The same material has a similar shape at different scan rates, but as the scan rate increases, the shifting of peaks in the figure becomes more significant, and the degree of shift becomes larger. After inspection, it was found that the reason why the 50 mv s-1 curve was different from the other curves was due to the misuse of the data measured by other experiments, which resulted in an error in the drawn image. Therefore, in the revision, the data or curves from 50 mV/s have been removed from the manuscript.

  1. Why the authors did not study the window potential of each electrode, but did it with the NTNS-NCW //NCW device (anode and cathode running together)?

Response: Thanks for your valuable suggestion about this issue. In this paper, three-electrode voltammetry and galvanostatic charge-discharge tests were performed on both electrodes. The results are shown in Figure 3 of the original text. Figure 3a and 3b are the CV and GCD curves of NCW, respectively; Figure 3c and 3d are the CV and GCD curves of NCNS-NCW. The electrodes were then assembled into devices with two electrodes, and two-electrode voltammetry and galvanostatic charge-discharge tests were also performed and the results were shown in Figure 4.

  1. Figures 4a, 4c, please update the X-axis. In this case, the authors did not use the reference electrode, so it should not have “vs. Hg/HgO, V”

Response: Thanks for your valuable suggestion about the manuscript. The wrong writing of the abscissa in the picture has been modified in Figure 4 in the original text.

  1. Please zoom in on the foot of EIS, we need to be able to see it clearly.

Response: Thanks for your valuable suggestion about the manuscript. The author for the EIS test in this paper has graduated and left our laboratory. At present, we only have the image of this EIS, which cann’t be zoom. However, the corresponding discussion on EIS has been performed in the manuscript.

As can be seen from the Figure S2 in Supplementary Information, EIS measurements were further performed on the NCW@NiCo2O4 composite electrode in the frequency range of 100 kHz-0.01 Hz. NCW can serve as a well-supported conductive substrate for NiCo2O4NSs. Therefore, the low Rs value (2.57 Ω) and Rct value (1.3 Ω) of the NCW@NiCo2O4-2 electrode suggest that the electrode has lower bulk resistance, faster electron transfer kinetics, and lower charge diffusion resistance. Rs value means series resistance and Rct value means charge transfer resistance. The lower resistance of NCW@NiCo2O4-2 is attributed to the three-dimensional porous interconnected conductive structure, which provides more exposed active centers and fast transfer paths for ions and electrons.

  1. Please measure the EIS of NCW as well as NTNS-NCW.

Response: Thanks for your valuable suggestion about the manuscript. we have added the electrochemical impedance spectra of NCW@NiCo2O4 composite electrodes as shown in Figure S2. And the EIS spectrum is explained in this paper. Because the author for EIS measurement has graduated and left the laboratory, the EIS of NCW cannot be supplemented with the test.

  1. It was introduced in the introduction that the material can be stable up to 10000 cycles, but we did not see the data about this, please provide it. Also please try to give the reason why the stability of the NTNS-NCW //NCW is smaller, which is only 7000 cycles.

Response: Thanks for your valuable suggestion about the manuscript. In the revision, we have added the cycling test till 15,000 cycles as shown in Figure S5. Figure S5 is placed in supplementary information.

Round 2

Reviewer 2 Report

I recommend this article to be published in the Polymers journal in presented form. 

Author Response

Thank you very much for your support and the recommendation of publication in present form.

Reviewer 3 Report

1.     Comment #4. The last digit of a scale is an error number, it cannot be trusted, not to mention that the result weight is only 0.01 mg. Your answer is not satisfied. If possible, please find a more detail scale to check your sample.

2.     Comment #5, 13. Please modify your main figure, these data is important. Thus, it must be in your main manuscript.

3.     Please edit your figure 3 and figure 4 more careful, it looked like they are cut from somewhere.

4.     Comment #12. If possible, please try to make the sample again for the EIS data, the foot of an EIS is very important.

For these reasons, we recommend the manuscript to be accepted after minor revision.

Author Response

Response to reviewer #3’s comments

Dear Editors and Reviewers,

Thank you for your letter and for the comments from the reviewers concerning our manuscript entitled “Facile electrodeposition of NiCo2O4 nanosheets on porous carbonized wood for wood-derived asymmetric supercapacitors”. We have carefully studied the opinions of the reviewers and revised the manuscript accordingly. These corrections are highlighted with a red color in the revised manuscript for review.

Reviewer #3:

  1. Comment #4. The last digit of a scale is an error number, it cannot be trusted, not to mention that Suggestions for Authors the result weight is only 0.01 mg. Your answer is not satisfied. If possible, please find a more detail scale to check your sample.

Response: Thank you for your valuable comments on the manuscript. Table S1 summarizes the mass of the deposited nanosheets. Samples with different loads can be obtained by controlling different deposition time. In addition, the quality of NCW and deposited nanosheets was measured with a microbalance (Sartorius BT125D) to ensure the accuracy of the data. The test results showed that the average weight of NCW was 10.0 mg, and the average weight of deposited nanosheets was 1.0, 2.0 and 3.0 mg, respectively. See Table S1 for specific loads.

Tab. S1 Mass loading of active material.

Sample

NCW (mg)

NiCo2O4 NSs (mg)

NCW

10.0

0

NCW@NiCo2O4-1

10.0

1.0

NCW@NiCo2O4-2

10.0

2.0

NCW@NiCo2O4-3

10.0

3.0

The mass of the deposited NiCo2O4 nanosheets was measured from the mass difference before and electrochemical deposition after post-heating treatment. The test results showed that the average weight of NCW was 10.0 mg, and the average weight of deposited nanosheets was 1.0, 2.0 and 3.0 mg, respectively. The NCW and NCNS-NCW materials directly utilized as self-standing electrodes after washing with alcohol under supersonic treatment.

  1. Comment #5, 13. Please modify your main figure, these data is important. Thus, it must be in your main manuscript.

Response: Thanks for your valuable suggestion about the manuscript. We have modified Figure 1 and added the EDS elemental mapping images to the main manuscript.

Fig. 1g showed the EDS elemental mapping images of the NCW@NiCo2O4-2, indicating the existence of C, N, O, Co, Ni elements in the samples.

  1. Please edit your figure 3 and figure 4 more careful, it looked like they are cut from somewhere.

Response: Thanks for your valuable suggestion about the manuscript. We have modified Figure 3 and Figure 4 with better resolution and readability.

  1. Comment #12. If possible, please try to make the sample again for the ElS data, the foot of an EIS is very important.

Response: Thank you for your comments. Unfortunately, the experiment designer and practical operator of this project have graduated, and the lower grade students involved in the experiment have also graduated. Therefore, at present, we are afraid that such EIS figure, but in the main manuscript, the corresponding data on Rct and Rs of samples 
